# Spatial and Temporal Evolution and Prediction of the Coordination Level of “Production-Living-Ecological” Function Coupling in the Yellow River Basin, China

**DOI:** 10.3390/ijerph192114530

**Published:** 2022-11-05

**Authors:** Yunhui Zhang, Zhong Wang, Shougeng Hu, Ziying Song, Xiaoguang Cui, Dennis Afriyie

**Affiliations:** 1School of Public Administration, China University of Geosciences (Wuhan), Wuhan 430074, China; 2Key Laboratory of Rule of Law Research, Ministry of Natural Resources, Wuhan 430074, China

**Keywords:** yellow river basin, production-living-ecological function, coupling coordination, spatial Dubin model, ARIMA model

## Abstract

To clarify the evolution of “production-living-ecological” function coupling in the Yellow River Basin, coordinating the spatial allocation of resources, development management and layout optimization, is an important means for achieving ecological protection and high-quality development in the region. In this paper, we conducted an empirical analysis and ARIMA prediction of the coupled production-living-ecological function coordination level in the Yellow River Basin of China from 2008 to 2018, and found that: (1) In terms of temporal evolution, the production-living-ecological function and coupling coordination level of each province and region in the Yellow River Basin generally shows a sharp and then slow upward trend, with the living functions changing more than the production and ecological functions; (2) in terms of spatial pattern, the production and living functions of each province and region show the trend of functional level increasing from east to west over time; the ecological functions, contrary to production and living functions, show a “high-low” aggregation, midstream shows “low-low” aggregation, and downstream shows “low-high” aggregation; (3) According to the regression results of the spatial Dubin model, the environmental governance level, technological research and development level, and social security level and resource dependence degree have positive promoting and spillover effects on the coupling coordination level of the “production-living-ecological” function in the region. However, population density and carbon emission intensity will hinder the development of regional coupling coordination level; (4) from the ARIMA prediction, the coupling coordination level of “production-living-ecological” in the Yellow River Basin continues the development trend of 2008–2018 in the short term, the overall coordination level is at a high level, and the variability of coupling coordination level among provinces and regions is further reduced. Finally, corresponding development countermeasures and suggestions are given to different provinces and regions based on the spatial and temporal evolution characteristics, influencing factors and development trend of the “production-living-ecological” function in the Yellow River basin.

## 1. Introduction

The Yellow River is one of the longest rivers in the world and the second largest river in China. Historically, the political, economic, and cultural center of China has long been located in the Yellow River basin. In recent years, rapid economic growth and rapid social transformation have resulted in a series of problems, such as the careless use of natural resources, unbalanced development of living space, and fragile ecological environment, which have seriously affected the sustainable development level of the basin itself and made the competition and contradiction of production, living, and ecological space in the basin increasingly intense. How to scientifically investigate the spatial and temporal evolution pattern of production-living-ecological in the Yellow River Basin, optimize the spatial functions of production-living-ecological based on its development trend, and promote the coordinated development of different spatial functions has become a pressing problem in the Yellow River Basin. It is urgent to solve those issues for the scientific management of national resources, ecological protection, and high-quality development.

Currently, much research has been conducted on the “production-living-ecological” function. These studies are based on land functions, with the value of ecosystem services [1,2,3], landscape ecology [4,5,6], land use change [7,8,9,10], and balanced development of land use benefits [11,12] as concerns. With the expansion of the research content of “production-living-ecological space”, the research of “production-living-ecological space” has been enriched under different research scales, such as national, provincial, municipal, and township. In terms of the “production-living-ecological” function of watersheds, the main studies include river continuity restoration [13,14], watershed hydropower utilization studies [15,16], hydrological productivity enhancement studies [17,18], biofuel impact studies on watershed water quality [19,20], integrated river basin management and maintenance [21,22,23,24], the quantitative identification of “production-living-ecological” functions of watersheds [25,26], and the sustainable development of ecological services in watersheds [27], among others. Some scholars have explored changes in ecological functions of watersheds in terms of natural factors and the effects of land use and land cover changes on hydrological conditions of watersheds, such as the frequency and severity of floods in watersheds, baseflow, flow, and degree of soil erosion [28]. Another important aspect is climatic factors, where climate change brings changes in precipitation, which in turn affects the water cycle [29], thus changing the flow path time, peak flow, flow volume, and sand production [30,31]. Shang Haiyang [32] used ecological compensation policy as an opportunity to assess the impact of ecological compensation policy on economic development in the basin. Feng Lichao [33] analyzed the development trend and characteristics of these three subsystems in the Yellow River Basin based on a comprehensive evaluation system of the ecological environment subsystem, urbanization subsystem, and high-quality economic subsystem and found that during the study period, the comprehensive level of the Yellow River Basin showed a significant upward trend; urbanization and high-quality economic growth story is higher than the ecological environment level. Meanwhile, urbanization and economic development show spatial differences in environmental carrying capacity; the degree of coupling has been at a high level and the coordination degree has changed significantly. Furthermore, it is inferred that the coordination degree of the river basin is steadily trending upward, shifting from disorder to order, and gradually reaching the level of coordination.

In recent years, with the national strategy of ecological protection and high-quality development of the Yellow River Basin, an increasing number of scholars have focused on the study of “production-living-ecological” based on the scale of the Yellow River Basin. The spatial identification and classification [34], function evaluation [35,36] and zoning optimization [37] are based on the “production-living-ecological” function. Lu Chengpeng [38], through a comprehensive assessment of the “production-living-ecological” function of 57 counties in the Gansu section of the Yellow River Basin, concluded that the “production-living-ecological” function index of the counties in the Gansu section of the Yellow River Basin showed an overall increasing trend. The spatial variation of the “production-living-ecological” function in the study area is dominated by the level of urban–rural integration; Zhang Yuzhen [39], supported by geospatial data and socio-economic statistics, identified the “production-living-ecological” function in the Yellow River Basin. The “production-living-ecological” function predominantly comprises two types: high production and low ecology and high ecology and low life. Wang Xiaofeng [40], based on land use data and geographic probes, discovered the differences in geographic orientation and the change patterns of production space, living space, and ecological space in the Loess Plateau region. He also found that socio-economic factors have a significant influence on the evolution of production-living-ecological space in the Yellow River Basin.

In summary, the research on “production-living-ecological” space in the Yellow River Basin focuses on the identification and assessment of factors affecting the “production-living-ecological space” in the Yellow River Basin, and the research methods are mainly based on social, economic, and environmental data to construct an index system or to build indicator systems or comprehensive indices using ArcGIS, geographic regression models, geographic probes, and other geographic analysis methods [41,42,43,44] to explore the spatial and temporal changes of “production-living-ecological” functions and driving factors. Although existing studies have initially investigated the function and functional zoning of the “production-living-ecological space” in the Yellow River Basin, the majority of these studies focus on a particular region or province in the basin, and relatively few studies have explored the evolution of the production-living-ecological space function in the Yellow River Basin from the perspective of the entire basin. Simultaneously, most of the existing studies concentrate on the current situation and the development trend of each spatial function in each region needs to be researched to a greater extent. Therefore, this paper constructs the index system of production-living-ecological function measurement in the Yellow River Basin, and on the basis of the evaluation results, it uses the coupled coordination model and the spatial Dubin model to explore the spatial characteristics and influencing factors of the level of production-living-ecological function and coupling coordination in the Yellow River Basin. The ARIMA model is used to predict the development trend of production-living-ecological spatial function in the Yellow River Basin. It helps to clarify the current status and development direction of the coupled production-living-ecological system in the Yellow River Basin and is beneficial for achieving the coordinated development of production-living-ecological functions in the Yellow River Basin and promoting their sustainability. It is important to accomplish the coordinated development of “production-living-ecological” functions in the Yellow River Basin and to promote the sustainable and high-quality development of the region.

## 2. Study Area Overview and Data Sources

The Yellow River Basin covers a total area of 795,000 km^2^ and flows through nine provinces (regions), covering more than 30% of the country’s population. It is China’s second largest river after the Yangtze River and is connected to the Qinghai-Tibet Plateau, Loess Plateau, North China Plain, and Bohai Sea natural ecological corridor, occupying an important position in China’s economic and social development and ecological security.

The Yellow River Basin is a major economic zone in China with a regional GDP of CNY 23.9 trillion at the end of 2018, accounting for about 26% of the country. In 2020, the Yellow River Basin coal production accounted for about 80% of the total national production, oil and natural gas production accounted for more than 30%, and new energy generation also accounted for 26%, which strongly supported national economic and social development, which shows that the economy of the Yellow River region depends on the rich natural resources. However, problems of the unreasonable industrial structure still exist in the basin. Some areas continue to rely on heavy industries, such as coal and oil as pillar industries, because they lack the resources to generate alternative energies. The Yellow River Basin still faces significant ecological and environmental issues, and the water quality of the basin and the air quality of each province and region is below the national average. At the same time, the Yellow River water resources are extremely limited, with only 2% of the country’s water resources to support a vital aspect of the people’s lives and industrial and agricultural production along the Yellow River provinces. The over-exploited water resources even lead to the interruption of the tributaries of the Yellow River.

This paper explores the development status of production function, living function, and ecological function in the Yellow River Basin from an economic, social, and ecological standpoint in nine provinces and regions, including Sichuan, Gansu, Ningxia, Inner Mongolia, Shaanxi, Shanxi, Henan, Shandong and Qinghai, comprising the study area based on the richness and dependence of resources of each province in the Yellow River Basin and the comprehensive consideration of data availability. Figure 1 shows the geographical location of the study area.The production function, living function, and ecological function data of the nine provinces and regions in the Yellow River Basin from 2008–2018 were obtained from the Chinese macroeconomic database, statistical yearbooks, and national economic and social development bulletins. Missing data were filled in by interpolation and extrapolation.

## 3. Index Construction and Research Methodology

### 3.1. Construction of “Production-Living-Ecological” Functional Index

In terms of index system construction, we refer to the perspective of production-living-ecological space index selection by related scholars [45,46,47] and combine the characteristics of rich coal resources, diverse ecological types, and economic construction in the Yellow River Basin to construct the evaluation index system of the “production-living-ecological” function (Table 1).

In terms of production function, the GDP per capita is selected to characterize the overall production level of the Yellow River Basin. The Yellow River Basin is an important agricultural producing area in China, accounting for 36 percent of the country’s grain growing area and 35 percent of its grain output. Meanwhile, the Yellow River basin is also an important base for energy, chemicals, raw materials, and basic industries in China, with more than half of the country’s coal reserves. Considering the Yellow River Basin’s rich agricultural and industrial resource reserves, this paper selects the per capita grain output and the land average output value of agriculture, forestry, animal husbandry and fishery in order to measure the regional agricultural production level. The output value of the coal mining industry and the sum of the profits of industries on the above scale are selected to reflect the regional industrial production level. The proportion of tertiary industry and energy consumption per unit output value are selected to characterize the regional industrial structure and the effect of industrial transformation.

In terms of living function, the Yellow River Basin is generally lower than the national average in terms of new urbanization and people’s living security, and there are still extensive, large, and deep poverty-stricken areas in the middle and upper reaches of the Yellow River Basin. Therefore, the unemployment rate is selected to measure the level of employment security in the Yellow River Basin; the urbanization rate, urban/rural Engel coefficient and urban–rural income ratio are selected to characterize the regional urban–rural development pattern; the total retail sales of consumer goods per capita are selected to reflect the improvement of people’s consumption level; and the number of beds in medical and health institutions per 1000 people and the road area per capita are selected to characterize the level of infrastructure construction.

In terms of ecological functions, the natural ecology of the Yellow River Basin is fragile and water resources are in short supply. At the same time, the resource utilization with high consumption and high emission and the unreasonable population expansion mode all pose a serious threat to the ecological security of the Yellow River Basin. Therefore, this paper selects the average water resources per capita and the average forest area per capita to characterize the environmental carrying capacity of the Yellow River Basin. The industrial wastewater emissions per unit of output value and sulfur dioxide emissions per unit of output value are selected to characterize the environmental pressure faced by the Yellow River Basin, and the nature reserves area, the urban sewage treatment rate, greening rate of built-up areas, and harmless treatment rate of domestic waste are selected to characterize the environmental protection efforts of the Yellow River Basin. In view of the serious problem of soil and water loss in the Yellow River Basin, the soil and water conservation area is also considered as an important indicator of ecological function.

### 3.2. Research Methodology

#### 3.2.1. Entropy Method

Entropy method is a method to determine the weight of indicators based on the size of the information load of each indicator, which can avoid the bias caused by subjective factors to a certain extent by analyzing the degree of connection between indicators and the amount of information provided by each indicator to determine the weight of indicators. Firstly, the research data need to be guaranteed dimensionless, so the polar difference method is used to standardize all indicators. Second, the indicators were assigned weights by the entropy method and the jth indicator of the ith program was calculated $X_{\mathrm{ij}}^{\prime }$ the weight p_ij_ and the credence entropy of the jth indicator; the coefficient of variation g_j_ of the jth indicator is then calculated and the weight matrix W_j_ is derived:(1)$$p_{\mathrm{ij}}=\frac{X_{\mathrm{ij}}^{\prime }}{\sum_{i=1}^{m}X_{\mathrm{ij}}^{\prime }}\left(i=1,2,\cdots ,m,\;j=1,2,\cdots ,n\right)\;$$
(2)$$e_{j}=-\frac{1}{\ln m}\overset{m}{\underset{i=1}{\sum }}p_{\mathrm{ij}}\ln p_{\mathrm{ij}};g_{j}=1-e_{j};W_{j}=\frac{1-e_{j}}{\sum_{j=1}^{n}1-e_{j}}\;$$

#### 3.2.2. Comprehensive Evaluation Method

After determining the weights of each indicator, it is possible to calculate the integrated development level of the “production-living-ecological space” function in the Yellow River Basin over the years (U_t_):(3)$$U_{t}=\overset{n}{\underset{i=1}{\sum }}W_{j}X_{ij}\;$$

In Equation (3), according to the different systems of indicators, the production function composite index U_1_, the living function composite index U_2_, and the ecological function composite index U_3_ of the Yellow River Basin can be calculated in turn.

### 3.3. Coupling Coordination Degree Model

The coupling degree is used to represent the interaction between two or more systems that closely cooperate and influence each other. By using the coupling degree function, we can reveal the inner synergistic mechanism of interaction and mutual influence among production system, living system, and ecosystem in Yellow River Basin. The production-living-ecological spatial function coupling degree function C of the Yellow River Basin:(4)$$C=3\times {\left[\frac{U_{1}\times U_{2}\times U_{3}}{{\left(U_{1}+U_{2}+U_{3}\right)}^{3}}\right]}^{1/3}\;$$

Similarly, the inter-functional coupling degrees C_1_, C_2_, and C_3_ between two functions can be obtained as:(5)$$\begin{array}{c} C_{1}=2\times {\left[\frac{U_{1}\times U_{2}}{{\left(U_{1}+U_{2}\right)}^{2}}\right]}^{1/2}\\ C_{2}=2\times {\left[\frac{U_{1}\times U_{3}}{{\left(U_{1}+U_{3}\right)}^{2}}\right]}^{1/2}\\ C_{3}=2\times {\left[\frac{U_{2}\times U_{3}}{{\left(U_{2}+U_{3}\right)}^{2}}\right]}^{1/2} \end{array}$$

The disadvantage of using the coupling degree function is that it can only describe the degree of coordinated development between systems and cannot determine whether the systems promote each other at a higher level or are closely linked at a lower level. Based on this, this paper further introduces the coupling coordination degree function, which not only reflects the degree of coordination between systems but also the stage of coordination development level, i.e., there are:(6)$$D=\sqrt{C\times T}\;;\;T={\alpha U}_{1}+{\beta U}_{2}+{\gamma U}_{3}\;$$

D denotes the coupling coordination degree function between production-living-ecological spatial functions in the Yellow River Basin. Similarly, the coupling coordination degree between two functions can be obtained as:(7)$$D=\sqrt{C\times T}\;;\;T={\alpha U}_{1}+{\beta U}_{2}\;\mathrm{or}\;T={\alpha U}_{1}+{\gamma U}_{3}\;\mathrm{or}\;T={\beta U}_{2}+{\gamma U}_{3}$$

α, β, and γ are the weights indicating the importance of production function, living function and ecological function, respectively. Referring to other scholars’ investigation on the interaction mechanism of production, living, and ecological spatial functions. In this paper, we believe that production, living, and ecological functions are equally important to the development level of any region, so α, β, and γ are taken as 1/3.

Based on the results of coupling coordination and with reference to existing studies [48,49], the coupling coordination level of “production-living-ecological functions” in the Yellow River Basin is classified as follows(Table 2).

### 3.4. Spatial Autocorrelation Analysis

Global spatial autocorrelation analysis was used to judge the overall spatial correlation of all the study samples. Calculating the global Moran index I:(8)$$I=\frac{n\cdot \sum_{i=1}^{n}\sum_{j=1}^{n}W_{\mathrm{ij}}\left(X_{i}-\bar{X}\right)\left(X_{j}-\bar{X}\right)}{\left(\sum_{i=1}^{n}\sum_{j=1}^{n}W_{\mathrm{ij}}\right)\cdot \sum_{i=1}^{n}{\left(X_{i}-\bar{X}\right)}^{2}}\;$$

As the formula shows, n is the total number of study samples; X_i_ and X_j_ represent the coupling coordination level of the “production-living-ecological” function of province i and j; and $\bar{X}$ is the mean of the coupling coordination level of province and region. Since the level of production, living, and ecological function is closely related to the economic development of each province or region, the spatial weight matrix W_ij_ is based on economic distance (the economic data used are the weighted average of per capita GDP from 2008 to 2018) is used to define the economic adjacency relationship of each province or region. The value range of I is (−1, 1), greater than 0 means positive correlation and closer to 1 means stronger spatial similarity of observations in different areas. A value less than 0 indicates a negative correlation; a value closer to −1 indicates a stronger spatial difference between the observed quantities in different regions; a value equal to 0 indicates that there is no spatial correlation between the observed quantities in different regions.

### 3.5. Spatial Durbin Model

The spatial panel model cannot only explain the attribute characteristics of a certain location but also quantitatively explain the spatial spillover effect of the surrounding location. With the increasingly close economic ties among provinces and regions, the indicators of resource dependence, government regulation, and scientific and technological research and development in this region will not only have an impact on the coupling coordination level of production-living-ecological functions in this region but also have an impact on the coupling coordination level of surrounding areas. In order to study the spatial effects of environmental governance, scientific, and technological research and development, resource dependence, resource allocation, social security, and the coupling coordination level of production-living-ecological function, this paper uses the spatial Dubin model (SDM) for empirical analysis. The empirical model is set as follows:(9)$$\begin{array}{cc} D_{\mathrm{it}}=\alpha +{\gamma \mathrm{Inw}}_{\mathrm{ij}}D & +\beta_{1}{\mathrm{EG}}_{\mathrm{it}}+\beta_{2}{\mathrm{TRD}}_{\mathrm{it}}+\beta_{3}{\mathrm{SRA}}_{\mathrm{it}}+\beta_{4}{\mathrm{SS}}_{\mathrm{it}}+\beta_{5}{\mathrm{RDD}}_{\mathrm{it}}+\beta_{6}{\mathrm{DP}}_{\mathrm{it}}+\beta_{7}{\mathrm{CEI}}_{\mathrm{it}}+\beta_{8}{\mathrm{RGC}}_{\mathrm{it}}+\delta_{1}W_{\mathrm{ij}}{\mathrm{EG}}_{\mathrm{it}}+\delta_{2}W_{\mathrm{ij}}{\mathrm{TRD}}_{\mathrm{it}}\\ & +\delta_{3}W_{\mathrm{ij}}{\mathrm{SRA}}_{\mathrm{it}}+\delta_{4}W_{\mathrm{ij}}{\mathrm{SS}}_{\mathrm{it}}+\delta_{5}W_{\mathrm{ij}}{\mathrm{RDD}}_{\mathrm{it}}+\delta_{6}W_{\mathrm{ij}}{\mathrm{DP}}_{\mathrm{it}}+\delta_{7}W_{\mathrm{ij}}{\mathrm{CEI}}_{\mathrm{it}}+\delta_{8}W_{\mathrm{ij}}{\mathrm{RGC}}_{\mathrm{it}}+\mu_{i}+\lambda_{t}+\varepsilon_{\mathrm{it}}\; \end{array}$$

D represents the level of coupling coordination; t represents the year; and i represents the region. Environmental governance investment reflects the level of environmental governance (EG); the number of patents applied per 10,000 people reflects the level of technical research and development (TRD); resource tax revenue reflects strength of resource allocation (SRA); and social security input reflects level of social security (SS). Social security level is used to reflect resource dependence degree (RDD). W_ij_ represents the weight matrix of economic geography; the control variables include the density of population (DP), carbon emission intensity (CEI); and resident Gini coefficient (RGC). μ_i_ represents the region fixed effect; λ_t_ represents time fixed effect; and ε_it_ represents the random disturbance term.

### 3.6. AMIRA Prediction Model

The autoregressive integrated moving average (ARIMA) model is introduced to predict the spatially coupled production-living-ecological functions in the Yellow River Basin, taking into account the temporal correlation and random fluctuation disturbances of the study subjects. The model is based on the autoregressive moving average model, in which the d-order difference is applied to the non-stationary time series so that the historical data can be used to predict future developments more accurately and, in the process, the optimal order *p* of the autoregressive term and the order q of the moving average term are determined. The expression of the model is as follows:(10)$$V_{t}=c+\phi_{1}V_{t-1}+\cdots +\phi_{p}V_{t-p}+\theta_{1}\varepsilon_{t-1}+\cdots +\theta_{q}\varepsilon_{q-1}+\varepsilon_{t}\;$$

In Equation (8), *c* denotes the constant term; *p* denotes the autoregressive order; q denotes the moving average term order; and $\;\varepsilon_{t}$ denotes the noise series of order t [50].

## 4. Spatial and Temporal Characteristics of the “Production-Living-Ecological” Function and Coupling Coordination Level in the Yellow River Basin

### 4.1. Time Evolution Trend

The comprehensive levels of the production function, living function, and ecological function of the Yellow River Basin provinces and regions from 2008 to 2018 were calculated according to Equations (3)–(5). As can be seen from Figure 2, the overall production function of the Yellow River Basin shows a gentle upward trend, increased by 120.21% over the 11 years, with a 68.81% growth in 2013 compared to 2008, 30.44% growth in 2018 compared to 2013. The average growth rate of the overall production function is 8.42%. The overall trend of the Yellow River Basin’s living function shows an obvious upward trend, with an increase of 170.67% in 2013 compared with 2008 and 43.63% in 2018 compared with 2013, and an overall increase of 288.75% in the basin’s living function over 11 years, with an average growth rate of 14.86%. The general trend of the Yellow River Basin’s ecological function shows a zigzag upward trend of rising, then falling, then rising again, with an increase of 63.62% in 2013 compared with 2008 and an increase of 37.03% in 2018 compared with 2013, with an overall increase of 124.22% and an average growth rate of 8.74% in ecological functions of the basin over 11 years.

Therefore, the production function, living function, and ecological function of all provinces and regions in the Yellow River Basin have increased more significantly over time, and the overall changes in production function, ecological function, and living function from 2008–2013 are greater than those from 2013–2018. The overall magnitude of change in the living function from 2008–2018 was greater than that of the production and ecological functions.

For each province and area in the Yellow River Basin, the coupling coordination degree of production-living-ecological function is computed based on the comprehensive index of production-living-ecological function. As shown in Figure 2, the overall coupling coordination level of production-living-ecological spatial functions in the Yellow River Basin was 0.1 in 2008, indicating a serious imbalance coordination level; 0.755 in 2013, indicating a moderate coordination level; and 0.995 in 2018, indicating a high coordination level. Between 2008 and 2018, the coupling coordination in the Yellow River Basin was represented by four stages: serious imbalance, basic coordination, moderate coordination, and high coordination, showing a more obvious upward trend overall.

### 4.2. Spatial Distribution Characteristics

ArcGIS 10.2 was used to visualize the integrated level of the production function, integrated level of living function, and the integrated level of ecological function in the Yellow River Basin, which was divided into five levels, and its spatial differentiation is shown in Figure 3. Figure 4 shows the proportion of the number of provinces and regions with different functional levels in all provinces and regions in different years, which reflects the evolutionary trend of functional proportions in different regions of the Yellow River Basin.

#### 4.2.1. Production Function

As shown in Figure 3 and Figure 4, the proportion of provinces and regions with lower and lowest production function decreased from 100% in 2008 to 67% between 2008 and 2018, while the proportion of provinces and regions with higher and highest production function increased from 0% to 11%, the proportion of provinces and regions with medium production function increased from 0% to 22%, and the overall production function level of the basin changed from low to high. In terms of spatial differences in production function among provinces and regions, in 2008, the production function of the lower reaches of the Yellow River (Shandong, Henan and Shanxi) was significantly higher than that of the middle reaches of the Yellow River (Shaanxi and Ningxia) and the upper reaches of the Yellow River (Qinghai, Sichuan, Gansu and Inner Mongolia), showing an obvious spatial pattern of “high in the east and low in the west”. In 2013, the production function of the middle reaches of the Yellow River (Shaanxi and Ningxia), Inner Mongolia, and Sichuan near the middle reaches began to gradually improve, from low values to lower values. In 2018, the production level of Inner Mongolia in the northern part of the basin and Sichuan in the southwest improved further, while the development of the production function of Shaanxi, Shanxi, and Ningxia in the middle reaches of the basin remained stagnant, forming a decreasing trend from the eastern and northern parts of the basin to the central and western parts.

According to the comprehensive evaluation results, the function of the production space in the Yellow River Basin is primarily determined by the level of agricultural production and industrial structure. The Shandong and Henan provinces, as developed agricultural regions in China, lay the foundation for the production of agriculture with their superior natural conditions located in the plains. In 2018, the average land value of agriculture, forestry, animal husbandry, and fishery in the Shandong and Henan provinces was 16 and 12 times that of the Gansu province, respectively. Meanwhile, Shandong, Henan, and Shanxi are rich in coal resources, which is another important reason why the production function of these provinces was significantly higher than that of other provinces in the early years. In 2008, the output value of coal mining industry in the three provinces accounted for 68.9% of the output value of coal mining industry in the Yellow River Basin, 50.4% in 2013, and 49.94 % in 2018, occupying an important geographical position in coal mining. Coal mining is also one of the channels for the significant improvement of Inner Mongolia’s production function. The output value of coal mining in Inner Mongolia increased by 217.54% from 2008 to 2018.

However, the production function of Inner Mongolia and Sichuan were enhanced by the increase in the total profit of industrial enterprises, which, over 11 years, increased by 82% in Inner Mongolia and by 207% in the Sichuan province, reflecting positive changes in its industrial structure and distinctive results in the industrial sector in terms of de-stocking, deleveraging, and cost reduction.

#### 4.2.2. Living Function

As shown in Figure 3 and Figure 4, from 2008 to 2018, the proportion of provinces and regions with lower and lowest living function decreased from 89% to 11%, the proportion of provinces and regions with higher and highest living function increased from 0% to 22%, the proportion of provinces and regions with medium living function increased from 11% to 67%, and the overall living function of the basin changed from low to high. In 2008, the living function of provinces and regions in the north and east was greater than that in the central and west, and from 2013, the living function of provinces and regions located in the central and west of the basin began to be generally improved; by 2018, the living capacity of the central and eastern regions was further improved and the difference in living function between provinces and regions began to be gradually reduced, but the living function in the western provinces (Qinghai) remains at a lower level, forming a “low-high” aggregation characteristic with the adjacent provinces.

The improvement of living functions depends on many factors, including the improvement of infrastructure, the improvement of residents’ living quality and the rationalization of urban and rural pattern. The per capita road area of the Shandong Province is significantly higher than that of neighboring provinces, which is 58.04% higher than the average level of the Yellow River Basin in 2008, 61.53% higher than the average level in 2013, and 35.77% higher than the average level in 2018, which reflects that Shandong has better traffic accessibility and provides more convenience for residents to travel. The high level of living function in the Shandong Province also depends on the significantly higher total retail sales of social consumer goods per capita than other provinces, which reached CNY 33,605 per person in 2018, higher than Inner Mongolia, which ranks second at CNY 13,010, and 40 times higher than Qinghai Province, reflecting to a certain extent the consumption level of the people and their need for a better material life. As for the middle and upper reaches of the Yangtze River provinces, such as Shaanxi, Shanxi, and Gansu, the gradual improvement of living functions is primarily attributable to the narrowing of the urban–rural gap, where the urban–rural income ratio gradually decreased from 4.1, 3.2, and 4.03 in 2008 to 2.97, 2.64, and 3.4 in 2018.

#### 4.2.3. Ecological Function

From Figure 3 and Figure 4, it is evident that from 2008 to 2018, that the proportion of provinces and regions with lower and lowest living function decreased from 89% to 78% and the proportion of provinces and regions with higher and highest ecological function increased from 0% to 11%, The proportion of provinces with moderate ecological functions rose from 66% to 77% and then fell back to 66%, while the overall change of the basin’s ecological function was less obvious and remained at a lower level. From the difference of ecological function of each province, the ecological level of the middle reaches of the Yellow River (Ningxia) and the upper reaches of the Yellow River (Inner Mongolia) gradually increases, showing the characteristics of low in the west and high in the east, and gradually formed the “low-high” aggregation pattern with Shandong and Henan as the low value and the “high-low” aggregation pattern with Qinghai as the high value.

Ecological function is mainly reflected in ecological carrying capacity, including regional water resources and forest area, which, to a certain extent, also reflects the strength of environmental protection and ecological restoration in the changed regions. The ecological function of Qinghai Province and Inner Mongolia Autonomous Region has been at a high value due to its rich forest resources and water resources. Ningxia has continued to rise in ecological function since 2008, also stemming from its vigorous protection of natural ecology, with the growth rate of 31.41% for the area under soil and water conservation. Inner Mongolia focuses on industrial pollution prevention and control, and its sulfur dioxide emissions per unit of output value dropped from 148.13 tons per billion CNY in 2008 to 17.9 tons per billion CNY in 2018, resulting in a more significant increase in its regional ecological function.

ArcGIS 10.2 was used to visualize the results of the level of coupled production-living-ecological spatial function coordination in the Yellow River Basin, which was divided into five levels: severe disorder, moderate disorder, basic coordination, moderate coordination, and high coordination. The spatial differentiation is shown in Figure 4.

### 4.3. Production-Living-Ecological Coupling Coordination

In the Figure 4 and Figure 5, from 2008 to 2018, the proportion of provinces and regions with seriously and moderately unbalanced coupling coordination decreased from 78% in 2008 to 0%, the proportion of provinces and regions with moderate and high coupling coordination increased from 0% to 55%, and the proportion of provinces and regions with basically balanced coupling coordination increased from 22% to 44%, and the overall coupling coordination of the basin changed from low to high. In terms of the difference of coupling coordination level among provinces and regions, in 2008, the production-living-ecological function coordination level in the upper and middle reaches of the Yellow River was lower than that in the upper reaches, with the Shandong Province as the high value forming a “high-low” aggregation feature; in 2013, the coupling coordination level in the upper and middle reaches of the Yellow River generally improved but overall it was still lower than that in the upper reaches. However, the overall coupling coordination level is still lower than that of the upstream region, and the coupling coordination level of Inner Mongolia and Shandong provinces is significantly higher than that of the surrounding areas, with Henan Province as the high value forming a “low-high” aggregation feature; in 2018, the coupling coordination level of the midstream and downstream regions improved more obviously, with the upstream region forming a “low-low” aggregation with Gansu and Ningxia as low values and the midstream region forming a “low-high” aggregation with Shanxi and Henan as low values.

The decreasing disparities in the coupling and coordination of production-living-ecological functions between provinces and regions indicate that the development models of each province and region are becoming more scientific and can guarantee the living standard of residents and environmental protection while taking into account the production, so that the “production-living-ecological” functions all demonstrate good interaction and sustainable development. For example, Henan and Shandong still need to prioritize the development of ecological functions, while the Qinghai Province needs to focus on the improvement of both production and living functions.

## 5. Analysis of Influencing Factors on the Coupling Coordination Level of the “Production-Living-Ecological” Function in the Yellow River Basin

### 5.1. Spatial Correlation Test

Table 3 shows that from 2008 to 2018, the Moran’s I index of the coupling coordination level of “production-living-ecological” in the Yellow River Basin is greater than zero, showing positive spatial correlation and passing the normal distribution test at a 1% level. This indicates that there is a spatial autocorrelation of economic distance between the “production-living-ecology” level of each province in the Yellow River Basin. The development of all provinces and regions is interrelated, and the provinces with better development of coupling coordination level will drive the coupling coordination level of neighboring provinces and regions.

### 5.2. Spatial Panel Model Test

Before the empirical analysis, the spatial Dubin model should be verified (Table 4). Firstly, the statistic of the Hausman test is 80.36, which passes the significance test of 5%, indicating that the fixed effect is better than the random effect. Secondly, the spatial panel model was selected. The statistics of LR Spatial Lag and Wald Spatial Lag tests were 37.85 and 36.02, respectively. The statistics of LR Spatial Error and Wald Spatial Error test were 44.67 and 52.72 and those of LM Spatial Lag and LM Spatial Error test were 33.09 and 6.09, respectively. All of these pass the significance test of 1%, which indicates that the analysis result of the spatial Dubin model is the best. Therefore, this paper will use the fixed effects of the spatial Dubin model to carry out a step analysis.

### 5.3. Analysis of Regression Results Based on Spatial Dubin Model

Table 5 shows the regression results of the spatial panel Dubin modularity of the coordination level of production-living-ecological function coupling in the Yellow River Basin. As can be seen from Table 5, the goodness of fit of the model is greater than 80%, indicating that the model can better reflect the relationship between variables. Of them, the explanation variable environmental governance (EG), technological research and development (TRD), social security (SS), resource dependence degree (RDD), control variable population density (DP), and carbon intensity (CEI) passed the 10% significance level, showing that the more variable area production-living-ecological function coupling coordination level has a significant effect. Of the above, environmental governance, technological research and development, social security, and resource dependence play a positive impact on the level of regional coupling coordination, and the influence decreases in turn. Population density and carbon emission intensity play a negative role in the level of regional coupling coordination, and the negative effect of population density is greater than that of carbon intensity. At the same time, the corresponding spatial effect coefficients of environmental governance, technological research and development, social security, and resource dependence are all positive and significant at the significance level of 5%, indicating that these variables have a significant role in promoting the coupling coordination level of the region and also promoting the improvement of the coupling coordination level of related regions.

### 5.4. Decomposition of Spatial Spillover Effect

In order to further investigate the influence in detail, this study uses partial differential method to decompose the total effect of the spatial Dubin model established in Section 5.3. Table 6 shows the regression results of direct effect, indirect effect, and total effect of SDM model.

Table 6 shows that the correlation coefficients of the direct effect, indirect effect, and total effect of environmental governance and technological research and development are all positive and pass the 1% significance level test. Of these, the direct effect coefficient of environmental governance is 0.036, the indirect effect coefficient is 0.052, and the total effect is 0.088, indicating that the environmental governance level of a region increases by 1%. It can improve the coupling coordination level of production, living, and ecology in this region by 0.036%, the coupling coordination level of related regions by 0.052%, and the coupling coordination level of watershed by 0.088%. The direct effect coefficient of scientific and technological research and development is 0.099, the indirect effect coefficient is 0.184, and the total effect is 0.283, indicating that a 1% increase in the level of scientific and technological R&D in a region can improve the production-living-ecological coupling coordination level of the region, the related region and the whole watershed by 0.099%, 0.184%, and 0.283%. The results showed that the provincial environmental governance and development level of science and technology not only can enhance the production of this region production-living-ecological function of the coupling coordination level, its spillover effects lead to the positive space, which also promote the associated region of the coupling coordination level, and through the positive feedback between region and region form the pattern of coordinated development. At the same time, the direct effect, indirect effect, and total effect of technological research and development are all higher than the correlation coefficient of environmental governance, indicating that technological research and development can improve the coupling coordination level of regional production-living space more significantly in the short term compared with environmental governance.

The correlation coefficients of the direct effect, indirect effect, and total effect of social security and resource dependence are all positive and pass the significance level test of 10%. However, the significance of the direct effect and indirect effect of the two are different, which reflects the existence of different degrees of positive spatial spillover effect. Of the above, social security has a strong impact on the coordination level of production-living-ecological coupling in the province but a weak impact on the related province, while the impact of resource dependence degree on the related province is stronger than that of the province. This different spatial effect also reflects the strong economic system and relatively weak social security system of the Yellow River Basin, which relies on industrial development.

Some variables also have negative spatial effects, which hinder the coupling coordination level of production-living-ecological functions in the Yellow River Basin to some extent. Of these, the total effect coefficient of population density is negative and passes the significance level test of 10%, indicating that excessive population density will indirectly cause a series of problems, such as resource allocation imbalance and social security pressure in the basin, and then affect the overall coupling and coordination function of the Yellow River Basin. The carbon intensity coefficient is negative and direct effect by 1% significance level test, shows for relying on the industrial production system in the development of river basin provinces and regions the extensive mode of production has a significantly negative effect on the basin ecological environment, which further highlights the environmental protection in the basin production-living-an important positive role in ecology.

## 6. Forecast of the Development of “Production-Living-Ecological” Function Coupling Coordination in the Yellow River Basin

The ARIMA model was used to predict the coupling coordination level of “production-living-ecological” function in provinces and regions of the Yellow River Basin. A number of potential alternative models were modeled and compared to select the best model and formula for each of them to form the predicted values for the next seven years (Table 7).

As can be seen from Table 7, the coupling coordination level of production-living-ecological in the Yellow River Basin continues the development trend of 2008–2018 in the short term, the overall coordination level is high, and the variability of the coupling coordination level among provinces and regions is further reduced. Specifically, the mean value of the coupling coordination level of each province and region over the next seven years is used as the ranking criterion for the coupling coordination level of each province and region, and the coupling coordination level of the upper reaches of the Yellow River Basin is greater than that of the lower reaches than that of the middle reaches. Among them, Inner Mongolia has the highest coupling coordination level, and the predicted value for the next seven years remains at the high coordination level with a mean value of 0.899; Henan has the relatively lowest coupling coordination level, and the predicted value for the next seven years changes from basic coordination to moderate coordination with a mean value of 0.661.

By comparing the coupling coordination level of “production-living-ecological” function of each province and region in the Yellow River Basin in each future year with the coupling coordination level of the previous year, the average growth rate of coupling coordination level of each province and region in the next seven years is obtained. In terms of growth rate, Gansu Province is predicted to have the fastest growth rate of 5.20% in the next 5 years, followed by Qinghai Province with 5.19%. The lowest growth rate is in Shandong Province with 2.41%, followed by Inner Mongolia with a 3.41% growth rate.

In order to verify the authenticity of the prediction results, the data from 2008 to 2013 are used as training data and the data from 2014 to 2018 are used as test data to verify the validity of the prediction results of ARIMA model. The test results are shown in Table 8.

Table 8 shows that from 2014 to 2018, the average relative error of AMIRA model prediction is 2.55%. Of these, the average relative error in 2014 is 3.34%, the average relative error in 2015 is 3.44%, the average relative error in 2016 is 2.57%, and the average relative error in 2017 is 1.74%, indicating that the overall prediction accuracy is high and the prediction results are credible.

## 7. Conclusions and Recommendations

### 7.1. Conclusions

This paper takes the Yellow River Basin provinces as the research object, constructs the evaluation index system of production function, living function, and ecological function as “production-living-ecological” function to measure the coupling coordination degree model. The spatial and temporal evolution and the driving factors of the coupling coordination level of “production-living-ecological” function in the Yellow River Basin was investigated using ArcGIS and the spatial Dubin model, and the ARIMA model was used to predict the future coordination level of “production-living-ecological” in the Yellow River Basin over the next seven years. The main findings are as follows.

In terms of temporal evolution, the overall level of production-living-ecological spatial functions and coupling coordination of the provinces and regions in the Yellow River Basin increased from 2008 to 2018 with the overall magnitude of changes in production, living, and ecological functions in each province and region being greater from 2008 to 2013 than from 2013 to 2018. Of these, the overall level and magnitude of change in living function is larger than the overall level of production and ecological function.

In terms of spatial patterns, from 2008 to 2018, there are obvious spatial differences in the overall coordination level of production function, living function, ecological function, and “production-living-ecological space” coupling in the Yellow River Basin. In terms of the spatial distribution characteristics of “production-living-ecological space”, the production function in the lower reaches of the Yellow River Basin is larger than that in the middle and upper reaches, and the spatial pattern shifts from “high in the east and low in the west” to “high in the east, high in the north and south, and low in the middle and west”. The spatial pattern of living function and production function is similar and increases from the east to the middle and west over time; the ecological function shows the opposite characteristics of production function and living function; that is, the upper reaches are larger than the middle and lower reaches and gradually changes to a diversified aggregation pattern, showing a “high-low” in general. More specifically, the upper reaches show a “high-low” aggregation, the middle reaches show a “low-low” aggregation, and the lower reaches show a “low-high” aggregation. The spatial characteristics of production-living-ecological function coupling and coordination in the Yellow River Basin are strongly correlated with the spatial characteristics of production, living, and ecological function zoning. Overall, a process of sequential upgrading from the upper reaches of the basin (northeast) to the middle and lower reaches (northwest) is shown.

According to the analysis of spatial influencing factors, it can be seen that the improvement of environmental governance, technological research and development, social security, and resource dependence have a significant positive effect on the coupling coordination level of production-living-ecological functions in the Yellow River Basin, while the population density and carbon emission intensity have a negative effect on the coordination level. By disaggregating the total effect, it is found that the direct effect and introduction effect of environmental governance and technological research and development in the provinces and regions of the Yellow River Basin are very significant, which is an important reason for the improvement of the coupling the coordination level of production-living-ecological function in the provinces and related provinces. The direct effect of social security is stronger than the indirect effect, and its effect on the improvement of coupling coordination level is more significantly reflected in the internal region. The indirect effect of resource dependence is stronger than the direct effect, and its effect on the improvement of coupling coordination level is more significantly reflected in the related region. The effect of population density on the production-living-ecological is hindered compared with the coupling coordination level of the production-living-ecological function in the whole watershed space. The hindering effect of carbon intensity is more relevant to the level of coupling and coordinated development of each province.

According to the ARIMA forecast, the coupling coordination level of production-living-ecological function in the Yellow River Basin will continue the development trend of 2008–2018 in the short term and the overall coordination level will be high, while the variability of the coupling coordination level among the provinces and regions will continue to reduce and each province and region will basically maintain a moderate coordination level. Of these, in the next seven years, the coupling coordination level in the upper reaches of the Yellow River Basin is greater than the level of the coupling coordination in the lower reaches and the middle reaches.

### 7.2. Recommendations

(1)Based on the spatial and temporal evolution rules and predicted trends of the “production-living-ecological” function coupling level in the Yellow River Basin, the following suggestions are made to improve the “production-living-ecological coupling level” in the Yellow River Basin. Each province and region in the Yellow River Basin should focus on the development of different spatial functions according to the local conditions and focus on the matching degree between different functions while enhancing their most disadvantaged functions to achieve scientific and sustainable development. For provinces and regions with lagging production functions, such as Qinghai Province, they should prioritize the development of highland special agriculture and animal husbandry such as yak, barley, and oilseed rape to reduce the economic reliance on pure grain production. In addition, those provinces also need to actively develop a highland tourism industry to promote the upgrading and transformation of industrial structures. For provinces and regions with lagging living functions, such as Qinghai Province and Gansu Province, they should promote the effective supply of basic public services in urban and rural areas and implement employment guarantees in order to narrow the income gap between urban and rural areas and improve the purchasing power and material and cultural living standards of residents. Northwestern provinces and regions with relatively lagging ecological functions, such as Ningxia, should strengthen the ecological restoration of nature reserves consolidating the prevention and control of desertification, implementing comprehensive control projects for soil erosion and water source conservation, improving regional ecological carrying capacity. For coastal areas of Shandong Province, the marine ecological environmental protection should be increased. Moreover, the tasks of precise pollution control, ecological restoration, risk prevention, the improvement of sea-friendly quality, and the development of marine carbon sink should be interwoven closely. For the central provinces, Shanxi and Henan, the effective prevention and management of industrial wastewater and industrial waste should be emphasized, and the reduction, resource utilization, and the safety of industrial waste should be encouraged.(2)From the perspective of spatial patterns, the coupling coordination level of provinces and regions in the Yellow River Basin has a significant spatial correlation; the Yellow River Basin should follow the model of coordinated regional development and actively integrate into the strategies of building “One Belt and One Road” and ecological protection and high-quality development of the Yellow River Basin. The neighboring provinces should focus on their spatial aggregation advantages in order to drive the development of the functionally disadvantaged areas by the functionally advantageous areas. At present, the central and western regions (the middle and upper reaches of the Yellow River) still face a situation where production advantages are not outstanding, living standards need to be enhanced, and the ecological environment needs to be protected. Considering the influencing factors of the coupling coordination level, all provinces and regions should pay more attention to the technological research and development and environmental protection. Against the background where the Yellow River Basin relies on primary resources for a long time, local governments need to further increase the advantages of cost externalities in the central and western regions, take advantage of technology and knowledge spillover in the transfer process, and improve their own technology level to reduce the pollution problem in industrial development. Simultaneously, the relatively abundant western natural resources (forest resources, water resources, coal resources, etc.) can be provided to the central and eastern regions, where natural resources are scarcer in order to improve resource allocation efficiency and alleviate regional development constraints.(3)According to the prediction results of coupling coordination degrees in future development processes, provinces and regions with moderate coordination, such as Gansu, Ningxia, Shaanxi, Shandong, Henan, Shaanxi, Sichuan, and Qinghai provinces and regions, should focus more on the inferior functions of a certain space, improve upon their strengths and complement their weaknesses, and focus on overall development level. Provinces and regions with high coordination, such as Inner Mongolia and Sichuan, need to maintain a high level of coupling coordination between different spatial functions and adopt a relatively steady development strategy. For other moderately coordinated provinces, inferior aspects should be increased according to the current situation of regional spatial function development in order to meet the requirements of regional sustainable development.

## Figures and Tables

**Figure 1 ijerph-19-14530-f001:**
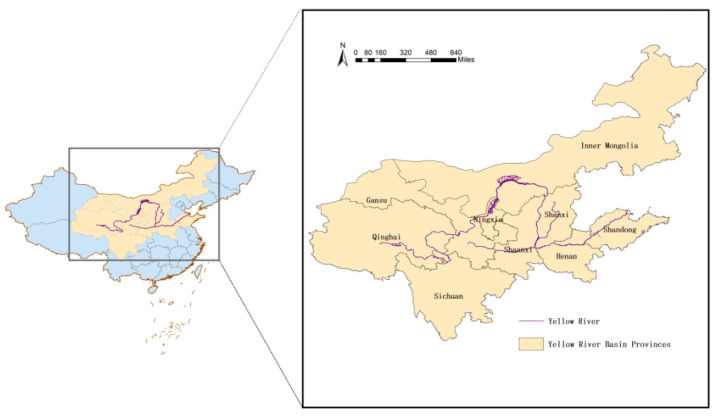
Study area.

**Figure 2 ijerph-19-14530-f002:**
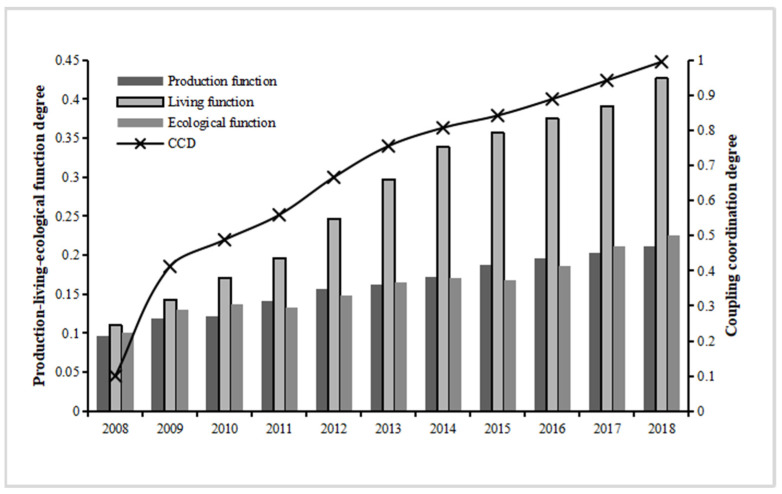
Time series evolution of “production-living-ecological” function coupling and coordination level in the Yellow River Basin.

**Figure 3 ijerph-19-14530-f003:**
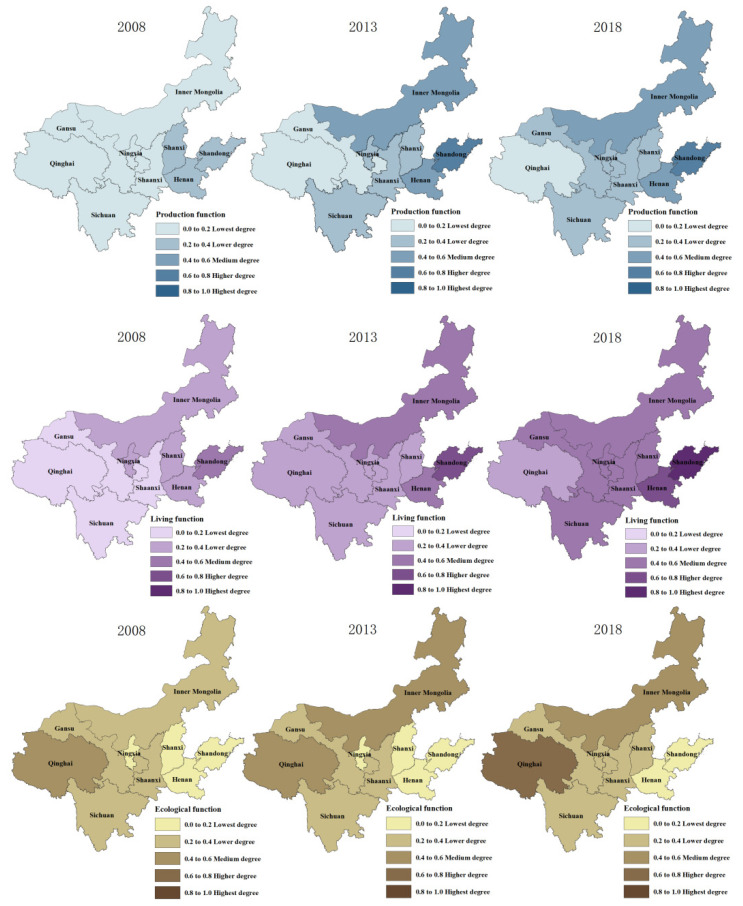
Spatial differentiation of “production-living-ecological” function in the Yellow River Basin.

**Figure 4 ijerph-19-14530-f004:**
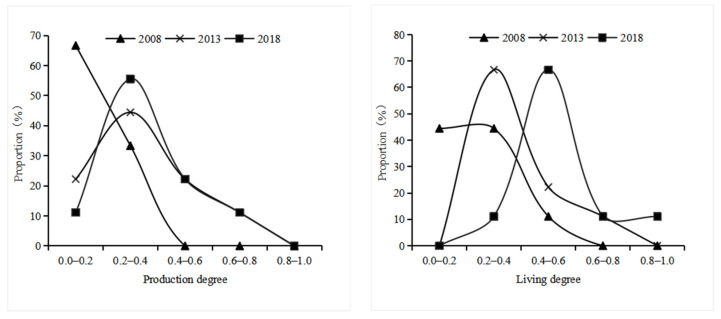
Evolutionary trend of the functional proportion of different regions in the Yellow River Basin.

**Figure 5 ijerph-19-14530-f005:**
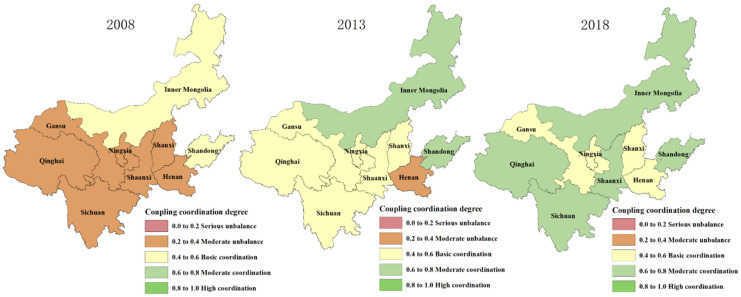
Spatial differentiation of the level of coordination of “production-living-ecological” spatial function coupling in the Yellow River Basin.

**Table 1 ijerph-19-14530-t001:** Functional evaluation index system of “production-living-ecological” function in the Yellow River Basin.

Criteria	First Level Indicator	Basic Level Indicator	Unit	Attributes	Weights
Production function	Overall production level	GDP per capita (X1)	CNY/person	+	0.0751
	Agricultural production function	Food production per capita (X2)	Kg/person	+	0.1089
		Average land value of agricultural, forestry, animal husbandry and fishery production (X3)	Million CNY/square kilometer	+	0.2491
	Non-agricultural production function	Fiscal revenue as a share of GDP (X4)	/	+	0.0577
		The proportion of tertiary industry (X5)	Kilometers/square kilometers	+	0.0684
		Energy consumption per unit of output (X6)	Killion tons of standard coal/billion CNY	−	0.0331
		coal mining industry output (X7)	Billion CNY	+	0.1898
		Total profit of industries above the scale (X8)	Billion CNY	+	0.0684
Living function	Employment security	Unemployment rate (X9)	%	−	0.0845
	Urban–rural development pattern	Urbanization rate (X10)	%	+	0.0953
		Urban–rural income level ratio (X11)	/	−	0.0755
		Urban Engel coefficient (X12)	/	−	0.0548
		Rural Engel coefficient (X13)	/	−	0.0418
	Consumption level	Total retail sales of consumer goods per capita (X14)	10,000 CNY/person	+	0.3478
	Infrastructure construction level	Number of beds per 1000 people in health care facilities (X15)	One bed/1000 people	+	0.1000
		Road area per capita (X16)	Square meter/person	+	0.2003
Ecological function	Environmental carrying capacity	Forest area per capita (X17)	Thousands of hectares/10,000 people	+	0.2684
		Water resources per capita (X18)	Cubic meter/person	+	0.3453
	Environmental pressure	Sulfur dioxide emissions per unit of output value (X19)	Tons/billion	−	0.0143
		Industrial wastewater emissions per unit of output value (X20)	Million cubic meters/billion CNY	−	0.0231
	Environmental protection efforts	Nature reserves area account for the proportion of the region (X21)	%	+	0.1928
		Soil and water conservation area account for the proportion of the region (X22)	%	+	0.0729
		Urban sewage treatment rate (X23)	%	+	0.0287
		Greening rate of built-up area (X24)	%	+	0.0247
		Harmless disposal rate of domestic waste (X25)	%	+	0.0297

**Table 2 ijerph-19-14530-t002:** Classification of production-living-ecological spatial function coupling coordination degree types in the Yellow River Basin.

Coupling Coordination	Coupling Coordination Level	Features
D ∈ (0, 0.2]	Serious unbalance	Regional production function, living function, and ecological function levels are extremely mismatched; regional production-living-ecological spatial structure imbalance
D ∈ (0.2, 0.4]	Moderate unbalance	Regional production function, living function, and ecological function levels are not matched; regional production-living-ecological spatial structure is not balanced
D ∈ (0.4, 0.6]	Basic coordination	Basic matching of regional production function, living function, and ecological function levels; basic balance of regional production-living-ecological spatial structure
D ∈ (0.6, 0.8]	Moderate coordination	The regional production function, living function, and ecological function levels are well matched; the regional production-living-ecological spatial structure is more reasonable
D ∈ (0.8, 1.0]	High coordination	Regional production function, living function, and ecological function are well matched; regional production-living-ecological spatial structure is scientific and reasonable

**Table 3 ijerph-19-14530-t003:** Moran’s I index of the coordination level of production-living-ecological function coupling.

Year	Moran’s I	*p*-Value	Year	Moran’s I	*p*-Value
2008	0.372 ***	0.002	2014	0.390 ***	0.001
2009	0.381 ***	0.001	2015	0.421 ***	0.001
2010	0.403 ***	0.001	2016	0.414 ***	0.001
2011	0.378 ***	0.002	2017	0.372 ***	0.001
2012	0.392 ***	0.001	2018	0.301 ***	0.001
2013	0.394 ***	0.001			

Note: *** is significant at 1% level.

**Table 4 ijerph-19-14530-t004:** Spatial panel model test results.

Test Items	Statistic	*p*-Value
Hausman Test	15.700 **	0.046
LM Spatial Lag	16.640 ***	0.002
LM Spatial Error	9.282 ***	0.000
LR Spatial Lag	36.98 ***	0.000
LR Spatial Error	46.55 ***	0.000
Wald Spatial Lag	54.65 ***	0.000
Wald Spatial Error	47.50 ***	0.000

Note: *** and ** are significant at 1% and 5% levels respectively.

**Table 5 ijerph-19-14530-t005:** Regression results of the spatial Dubin model.

Variable	Coefficient	*p*-Value
lnEG	0.035 ***	0.000
lnTRD	0.095 ***	0.000
lnSRA	0.006	0.343
lnSS	0.056 ***	0.000
lnRDD	0.020 **	0.045
lnDP	−0.021 *	0.099
lnCEI	−0.017 ***	0.000
lnRGC	−0.027	0.271
WlnEG	0.058 **	0.023
WlnTRD	0.202 ***	0.000
WlnSRA	0.001	0.962
WlnSS	0.102 **	0.041
WlnRDD	0.071 **	0.010
WlnDP	−0.035	0.172
WlnCEI	−0.011	0.433
WlnRGC	−0.037	0.683
R-squared	0.9013	
log-likelihood	229.315	

Note: ***, ** and * are significant at 1%, 5% and 10% levels respectively.

**Table 6 ijerph-19-14530-t006:** Total effect decomposition of the spatial Dubin regression model.

Variable	Direct Effect		Indirect Effect		Total Effect	
	Coefficient	*p*-Value	Coefficient	*p*-Value	Coefficient	*p*-Value
lnEG	0.036 ***	0.000	0.052 ***	0.002	0.088 ***	0.000
lnTRD	0.099 ***	0.000	0.184 ***	0.000	0.283 ***	0.000
lnSRA	0.006	0.364	0.001	0.969	0.007	0.729
lnSS	0.059 ***	0.000	0.097 *	0.075	0.156 ***	0.010
lnRDD	0.020 *	0.075	0.063 **	0.014	0.084 ***	0.005
lnDP	−0.021	0.134	−0.029	0.155	−0.051*	0.099
lnCEI	−0.017 ***	0.000	−0.009	0.468	−0.027	0.101
lnRGC	−0.034	0.229	−0.049	0.572	−0.008	0.444

Note: ***, ** and * are significant at 1%, 5% and 10% levels respectively.

**Table 7 ijerph-19-14530-t007:** ARIMA prediction results.

Watershed	Provincial Areas	Optimal Model	Backward 1 Year	Backward 3 Years	Backward 5 Years	Backward 7 Years	Average Value	Coordination Type
Upstream	Gansu	ARIMA(0,1,0)	0.622	0.694	0.766	0.837	0.729	Moderate coordination
	Sichuan	ARIMA(1,1,0)	0.740	0.820	0.905	0.992	0.864	High coordination
	Ningxia	ARIMA(1,1,0)	0.583	0.643	0.703	0.764	0.673	Moderate coordination
	Inner Mongolia	ARIMA(1,1,0)	0.810	0.869	0.929	0.990	0.899	High coordination
	Qinghai	ARIMA(0,1,0)	0.637	0.710	0.783	0.856	0.747	Moderate coordination
Midstream	Shanxi	ARIMA(0,1,0)	0.608	0.654	0.701	0.748	0.678	Moderate coordination
	Shaanxi	ARIMA(0,1,1)	0.624	0.674	0.725	0.775	0.700	Moderate coordination
Downstream	Henan	ARIMA(0,1,0)	0.578	0.633	0.688	0.743	0.661	Moderate coordination
	Shandong	ARIMA(0,1,0)	0.704	0.740	0.776	0.758	0.758	Moderate coordination

**Table 8 ijerph-19-14530-t008:** Test of coupling coordination degree prediction results.

Provincial Areas	2014		2015		2016		2017		2018	
	True Value	Fitted Value	True Value	Fitted Value	True Value	Fitted Value	True Value	Fitted Value	True Value	Fitted Value
Gansu	0.510	0.521	0.526	0.546	0.544	0.562	0.564	0.580	0.587	0.599
Sichuan	0.611	0.622	0.630	0.638	0.647	0.656	0.673	0.672	0.705	0.704
Ningxia	0.484	0.470	0.493	0.510	0.505	0.532	0.531	0.543	0.547	0.563
Inner Mongolia	0.741	0.778	0.729	0.759	0.736	0.738	0.759	0.755	0.783	0.785
Qinghai	0.52	0.509	0.512	0.556	0.515	0.548	0.571	0.551	0.601	0.607
Shanxi	0.531	0.535	0.539	0.554	0.550	0.562	0.555	0.573	0.585	0.578
Shaanxi	0.560	0.543	0.565	0.596	0.567	0.566	0.599	0.592	0.612	0.629
Henan	0.437	0.390	0.462	0.464	0.509	0.489	0.542	0.536	0.551	0.569
Shandong	0.642	0.644	0.658	0.659	0.674	0.675	0.685	0.691	0.687	0.702

## Data Availability

The datasets used and analyzed in the current study are available from the corresponding author on reasonable request.

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
