# Peer review of "Spatial and Temporal Evolution and Prediction of the Coordination Level of “Production-Living-Ecological” Function Coupling in the Yellow River Basin, China"

_ijerph, 2022, doi:10.3390/ijerph192114530_

Round 1
Reviewer 1 Report
The subject itself could be interesting if developed with more accuracy. The data that is used it is from other studies and it is rather generic, using just a few markers that offer only a general image for a area, hiding real problems. For a better understanding of each situation, I believe that more data is needed for each situation in particular regarding the specific element that define economy, living areas and ecology.
Another issue is that this study uses rather old data (2008-2018), a single prediction method, that does not take into account other parameters that influence the relations between those tree functions and the prediction is for a period of five years (four of them already passed). From my point of view key elements such as type of economical units, poverty, unemployment rate, occupancy for housing, level of happiness or the level of attractivity, changes in price for land acquisition for an area are elements that are influencing the relations between the analyzed functions.
Another defining element is the evolution of the society and their ideologies, the urban sprawl issue as well the ghost city issue that emerges in developing areas and has a huge impact on the ecological function. The number of natural habitat reservations and parks corelated with the designated area and virgin lands could improve the data regarding ecological aspects. The building materials used for new developments has a great impact on the ecological part as well and can provide data needed for a more accurate prediction.
In the end I must notice that the discussion part is missing and only a vague prediction table is provided. The conclusions are vague but offer accurate solutions for specific areas that are not argued nor linked to references.
Reviewer 2 Report
1. This article just analyze the superficial temporal ans spatial evolutionary features of living function, production function, and ecological function, but without deeply analyzing the potential driving forces.
2. The evaluation system can also be applied into other River Basin, why they are special to the Yellow River Basin.
3. The prediction of coupling coordination degree doesn't seem to make any sense, how to validate the prediction accuracy of this smooth trend method. The best prediction method should consider different situational interventions.
4. the map outline of Qinghai is different in your all figures. where is the functional share of different regions in Figure4.
Round 2
Reviewer 2 Report
thanks for your hard work. the outline of Qinghai still is wrong in your figure, please map it right.